# Non-Price Criteria for the Evaluation of the Tender Offers in Public Procurement of Ukraine

**Alexander Baranovsky** [1], **Nataliia Tkachenko** [1], **Vladimer Glonti** [2],
**Valentyna Levchenko** [3], **Kateryna Bogatyrova** [4], **Zaza Beridze** [2], **Larisa Belinskaja** [5]
**and Iryna Zelenitsa** [6,*]

1   Faculty of Banking Technology and Business, University of Banking, 04070 Kyiv, Ukraine;
    bai@ubs.edu.ua (A.B.); ivan99@ukr.net (N.T.)
2   Faculty of Economics and Business, Batumi Shota Rustaveli State University, 6010 Batumi, Georgia;
    vladimer.ghlonti@bsu.edu.ge (V.G.); beridzezaza@yahoo.com (Z.B.)
3   Faculty of Economics and Business, Kyiv National University of Technologies and Design,
    01011 Kyiv, Ukraine; valentyna.levchenko@gmail.com
4   Faculty of Trade and Marketing, Kyiv National University of Trade and Economics, 02156 Kyiv, Ukraine;
    bogatyriova@ukr.net
5   Faculty of Economics and Business Administration, Vilnius University, 01513 Vilnius, Lithuania;
    Larisa.belinskaja@evaf.vu.lt
6   Faculty of International Trade and Law, Kyiv National University of Trade and Economics,
    02156 Kyiv, Ukraine
*   Correspondence: zelenitsa@ukr.net; Tel.: +38-093-199-75-89

**Abstract:** Traditionally, public procurement has been associated with the measurement of achieving savings. However, recent research shows that the economic impact of public procurement is not limited only to savings, but by measuring the impact of four capitals—natural, human, social, and economic—on sustainable well-being over time. Ukraine is a country with a very low gross domestic product (GDP) per capita, which exacerbates the problem of the impact of public procurement results on the population's welfare. Ukrainian public procurement legislation allows customers to apply non-price criteria (the share of non-price criteria cannot be more than 70%), which, together, are taken into account in the formula of the quoted price. The studies show that the effect of the use of non-price criteria depends on the relevance of the method of the evaluation of non-price criteria. The most important non-price criteria for Ukrainian customers by product categories and the methods of their evaluation are analyzed according to the Bi.prozorro.org analytics module. Therefore, it is concluded that the quoted price method, which is used in Ukrainian practice, is not relevant in comparison with the method used in the EU. A survey of the government buyers on the practice of applying non-price criteria was conducted, and the areas of their use were identified.

**Keywords:** public procurement; evaluation of tender offers; non-price criteria; life cycle cost; quoted price

## 1. Introduction

One of the most promising areas of e-commerce is so-called e-procurement (Gelderman et al. 2006), which is increasingly gaining ground in the public procurement markets of developed countries (SIGMA 2016), in particular the European Union common market. In fact, e-procurement is the flow and management in the electronic environment of all stages of the procurement cycle, including marketing research, determining the range of economic agents (potential suppliers), procurement procedures, placing orders, delivering and paying for them, budgeting, and planning procurement

(formation of budgeting procurement and its implementation) (Shatkovskyi and Faivesh 2015). Public sector procurement is increasingly seen as an important instrument for inducing innovation in the private sector (Lenderink et al. 2019). Sustainable public procurement is a powerful tool to reflect on national strategic intentions and promote scientific and technological innovation (Hu et al. 2018; Wang et al. 2018a, 2018b; Jobidon et al. 2018; Ivashova and Ivashov 2019, p. 327).

The important step in the public procurement process is to determine the winner with which the procurement contract is concluded. From a market standpoint, considering the specificity of the product, work or service, there is the possibility of its realization, and it is at the point of intersection of the interests of a buyer and a seller. Therefore, the determining factor when choosing a tender winner is an evaluation of its tender offers (Vaidya et al. 2006) on the basis of the proportion of price and non-price criteria, due to the specificity of the procurement items (product, work, or service) (Costantino et al. 2012), technical specifications (Crown Agents 2017a, p. 3), and future operating costs (Edquist et al. 2015).

Therefore, knowing and characterizing public procurement announcements (tenders) is fundamental for managing public resources well (Rodríguez et al. 2019).

The application of the additional (non-price) offer evaluation criteria that are essential to the buyer, such as payment terms, date for implementation, warranty service, etc., is important for both public and commercial procurements (Wozniak et al. 2018; Zimon et al. 2019).

The enactment of the law of Ukraine on public procurement in 2015 provided the opportunity for government buyers to apply non-price criteria to select cost-effective proposals. Environmentally friendly procurement maintenance, the quality of health care, improvement of medical equipment, access to quality education, trust maintenance in the government agencies, the efficient use of economic resources, the development of small and medium enterprises, innovation, and the stimulation of the competitiveness of enterprises are associated with public procurement. For example, if public procurement is considered as a means of the realization of environmental goals through the mechanism of the circular economy, it contributes to the use of resources during and after their life cycle with further processing, and the further use of procurement, and changes the country's environmental deficit. The Europe 2020 strategy has set an objective that public procurement should mitigate (OECD 2019) climate change, by reducing greenhouse gas emissions by 20%, and increasing energy efficiency by 20%. This approach adds new value to public procurement through its impact on the country's natural, human, social, and economic capital (Zagirniak et al. 2018; Komarova et al. 2019), which changes the well-being of the population over time. However, to ensure the value of public procurement, government officials need the following tools: an objective method of evaluation of non-price criteria and tenders; the presence of examples of successful procurement with the impact of non-price criteria on efficiency; the opportunity for supplier innovators to be competitive in tenders in the conditions of application of mass price dumping by traditional participants; and a high level of buyer's professionalism.

Different methods which are used to evaluate the same offers can give different results in determining the successful tender winner. The non-price criteria for the evaluation of tenders must be as objective and quantifiable as possible, and are important for conducting electronic public procurement. The neglect of non-price criteria that could reduce the cost of servicing goods in the future may have negative consequences for meeting public and state needs, which in turn reduces the effectiveness of using public services. This is especially notable during the construction of hospitals, schools, and roads, in providing the population with medications and more. The volume of public procurement in Ukraine in 2019 was UAH 1.03 trillion (at expected value), UAH 31.7 billion of which (3.1%) was announced with non-price criteria.

The electronic evaluation of tender offers is one of the stages that characterizes the level of development of electronic public procurement in a particular country. In 2015, 71% of the EU countries implemented this phase (Tkachenko 2016).

According to Draskovic et al. (2017, p. 596), the conditions of production, exchange, etc. (price regulation, investment decisions, scope of state purchases, changes in foreign trade conditions, etc.) are often crucial (specific interest) for certain groups of people.

Cholopray (2019) considers that the evaluation of tender offers should be an organized process for selecting the best offer to achieve this goal, and pays more attention to the research on how fair the price of tender offer is. The scientist believes that prices justification needs to be analyzed in order to make sure that the participant's price is fair. There is no such practice in Ukraine. The latest changes in Ukrainian public procurement legislation, from April 2020, will introduce the possibility of rejecting tender offers with abnormally low prices.

The main non-price criteria for the evaluation of tender offers are the term, the guarantee, the payment terms, the functional and environmental properties of the product (Duda 2016). Despite the feasibility of the application of non-price criteria, only 17% of the tender's non-price criteria have been applied in the Czech Republic. The author explains such a low proportion to the fact that customers are afraid of responsibility for being objective in the evaluation of the offers before the Bureau of Competition Protection based on the audit results of the Czech Republic. In Ukraine, state audit bodies monitor and review procurement only in compliance with legal requirements. The analysis of inflated purchase prices is actively carried out by public organizations that make such cases public.

The application of environmental (Walker and Brammer 2012; Koziuk et al. 2019) and social criteria (Khoma et al. 2018; Glonti et al. 2020) in the area of public procurement (Semple 2015) also takes place, but many customers do not take into consideration the financial sustainability of the participants, which is a more important criterion when choosing a supplier. We agree with the author's position and believe that such a practice should be implemented in Ukraine.

In the study of the evaluation of the tender offers by government customers, the scientists (Mateus et al. 2010) consider a linear scoring function and note the importance of openness of information on non-price criteria according to the Code of State Contracts. In Ukraine, information on the application of non-price criteria is open, and the price method is the only one that can be applied, which is a significant weakness (Tkachenko et al. 2020).

In public procurement guidelines, the EU project "Harmonization of the Public Procurement System in Ukraine with EU Standards" (Crown Agents 2017a, 2017b) the experts have stated that compliance with the technical specifications on a two-point scale "meets"/"does not meet" does not comply the principle of the best offer in terms of the quality. This two-point scale practice is used by the customers in many cases in Ukraine.

The use of non-price criteria in the evaluation of the tender offers allows one to choose the most cost-effective offer, but on the condition that such criteria are significant, the evaluation methodology is adequate and the evaluation results are controlled.

## 2. Aims and Methods

The aim of the study is to identify important non-price criteria for Ukrainian government buyers based on the ProZorro (2019) system analytics module; the assessment of the adequacy of the quoted price method which is used in the Ukrainian practice; customers survey on the awareness of the need to apply non-price criteria; the implementation of author's questionnaire of polling government customers at the online training "Electronic evaluation of tender offers proposals using non-price criteria."

The analysis of non-price criteria used by the tenderers was based on the ProZorro (2019) analytics module. Non-price criteria have been studied; product groups for which customers have defined non-price criteria and how the use of non-price criteria affect the level of competition have also been investigated. On the one hand, the use of non-price criteria should not restrict competition between suppliers, and, on the other hand, it should allow suppliers who offer the products with higher quality, safety, innovation, environmental measures to be competitive in public procurement by price criteria.

The assessment of the adequacy of the quoted price method used in the Ukrainian public procurement system has been performed on the basis of the comparison of the results using other

methods, such as the estimated value method and the comparative method. This comparison allows us to make conclusions on whether an objective choice of the best offer is achieved, taking into account non-price criteria.

The survey of the customers has been conducted in order to assess their level of awareness of the purpose of non-price criteria, and the ratio of expectations and results in the application of non-price criteria.

## 3. Results

According to the data of the professional analytics module, the use of non-price criteria for the evaluation of the tender offers in Ukraine in 2019 was observed in 1.9 thousand of the tender's procedures for the nomenclature and lots of the expected value of UAH 31.7 billion. Overall, 97% of public procurement procedures (at the expected value) are carried out under a single price criterion. For comparison, in the EU, the share of purchases for the price criterion is carried out only from 12 to 97% (Table 1).

**Table 1.** Proportion of procedures in which the contract award was carried out only on a price criterion, 2018, %.

| Country | % |
|---|---|
| Malta | 97 |
| Lithuania | 96 |
| Slovakia | 96 |
| Cyprus | 93 |
| Iceland | 92 |
| Greece | 90 |
| Estonia | 88 |
| Romania | 86 |
| Bulgaria | 84 |
| Sweden | 79 |
| Czech Republic | 77 |
| Latvia | 74 |
| Liechtenstein | 71 |
| Portugal | 69 |
| Luxembourg | 67 |
| Germany | 65 |
| Slovenia | 63 |
| Finland | 56 |
| Poland | 48 |
| Hungary | 44 |
| Denmark | 42 |
| Italy | 39 |
| Belgium | 37 |
| Croatia | 34 |
| Spain | 31 |
| Austria | 28 |
| Ireland | 24 |
| Norway | 24 |
| Netherlands | 18 |
| France | 12 |
| United Kingdom | 12 |

Source: compiled by the authors according to the data (European Commission 2019).

According to the ProZorro (2019) electronic procurement system analytics module, non-price criteria are used in 3.7% for the procurement of goods, 4.4% for services and 91.9% for works.

The main non-price criterion for the purchase of goods is the delay of payment that, in the percentage terms is 41% (Figure 1), including the term more than 180 days from the date of delivery of goods, receiving of services, execution of tenders, which is a negative phenomenon of the financing of public procurement.

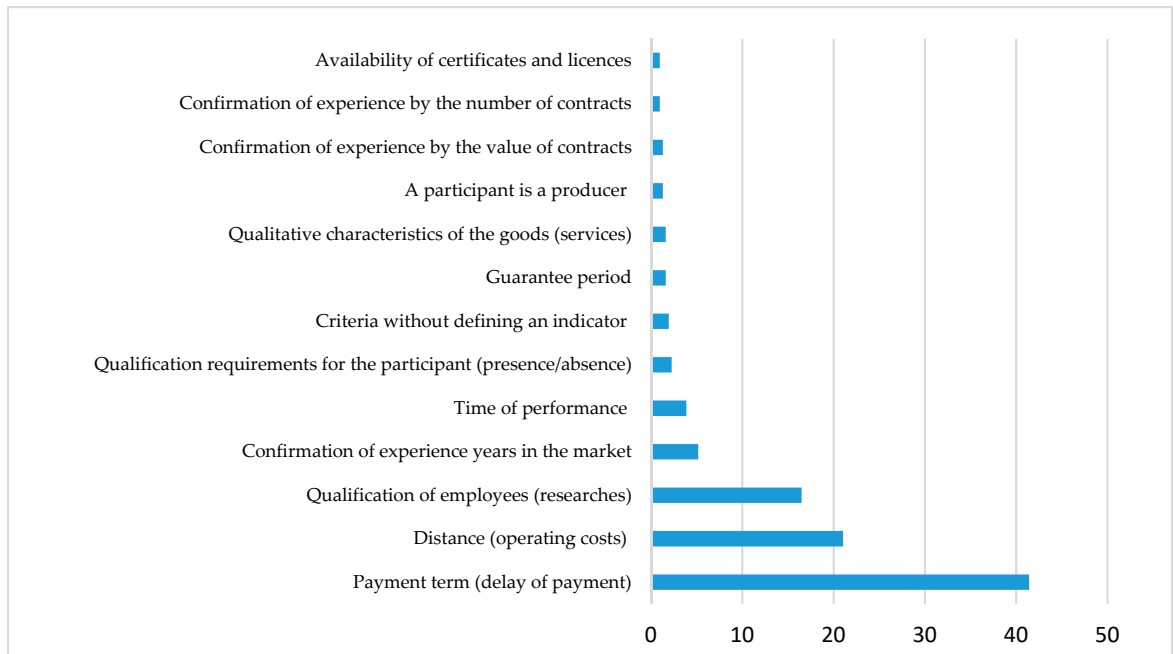

**Figure 1.** Structure of non-price criteria specified by the customer in the procurement of goods announcement in 2019. Source: compiled by the authors according to the data of the analytics module (Bi.prozorro 2019).

The law of Ukraine stipulates that non-price criteria can be used only in the case of complex or specialized procurement, and in the case of the procurement of goods, works and services produced, performed or provided not for a separately developed specification (technical design)—only the price is a permanent market for them. We have analyzed the nomenclature of goods purchased by non-price criteria (Figure 2). The studies showed that non-price criteria were used to purchase goods, which required the use of price criterion only by the law. No risk indicator is monitored (Merlo et al. 2013, p. 446; Xue et al. 2018; Pyrkova et al. 2018, p. 122; Kaigorodova et al. 2018; Glonti et al. 2019, p. 126; Vinogradova et al. 2019).

Non-price criteria have been observed in 14% of the purchases of goods with non-price criteria, which are determined for the selection of a specific winner—a sign of corruption by the customers. These are criteria, such as a non-payer of value added tax; the availability of certificates and licenses; experience of execution of similar contracts, experience of delivery for the budget sphere.

The study has shown that non-price criteria used by tenderers are absurd. For example, a delay of payment without charging penal sanctions (ProZorro 2018) indicates that the customer considers it as the best offer of the tenderer, who will not use penal sanctions, in case the customer fails to execute the contract in terms of timely payment.

We have investigated how the application of non-price criteria affected the level of competition (Table 2).

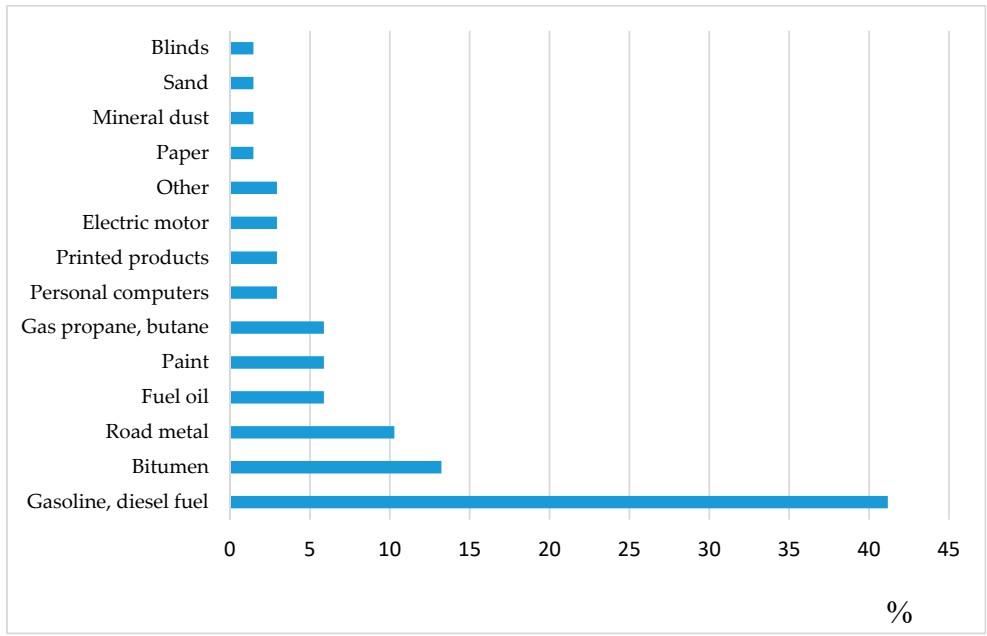

**Figure 2.** Goods purchased by customers by non-price criteria in 2019. Source: compiled by the authors according to the data of the analytics module (Bi.prozorro 2019).

**Table 2.** Level of competition for selected items, depending on the use of non-price criteria.

| No. | Group of Goods | The Level of Competition in Procurement | |
| --- | --- | --- | --- |
| | | Without Non-Price Criteria | With Non-Price Criteria |
| **Works** | | | |
| 1 | Construction of pipelines, roads | 2.40 | 3.05 |
| **Goods** | | | |
| 2 | Transport equipment | 2.59 | 2.78 |
| 3 | Petroleum products, fuel, electricity | 2.79 | 2.80 |
| 4 | Foodstuffs | 2.09 | 3.00 |
| 5 | Printed products | 2.62 | 2.00 |
| 6 | Office and computer appliances | 3.03 | 2.41 |
| 7 | Medical equipment and pharmaceutical products | 2.15 | 2.09 |
| **Services** | | | |
| 8 | Repair and maintenance services | 2.02 | 2.40 |
| 9 | Services in the fields of sanitation and environmental protection | 2.23 | 2.57 |
| 10 | Hotel, restaurant and retail services | 2.03 | 2.65 |

Source: compiled by the authors.

The analysis of the use of non-price criteria by the tender organizers has given the following results:

1. Ukrainian customers do not have the current practice of correctly applying non-price evaluation criteria for the tender offers. They often apply irrelevant and nonobjective non-price criteria for evaluating tenders, especially when purchasing goods, which leads to the violation of the principle of maximum procurement efficiency.
2. The use of non-price evaluation criteria for participants in most product groups has led to increased competition in the tenders.
3. The model of evaluation of tender offers with non-price criteria, which is regulated and applied in Ukraine, is unsuccessful for the selection of the best offer.

The national public procurement legislation uses the method of reducing the quoted price to evaluate the tenders with non-price criteria (Formula (1)).

$$PP = P/(1 + (F1 + F2 + \ldots Fn)/PV, \tag{1}$$

where

PP—the quoted price;
P—the offer price;
F—the value of non-price criterion of i-participant;
PV—the importance of "price" criterion.

or

$$PP = P/KK, \tag{2}$$

where

KK—the correction coefficient

$$KK = 1 + (F1 + F2 + \ldots Fn)/PV \tag{3}$$

Table 3 shows the initial data for the calculation of the quoted price.

**Table 3.** Initial data for the calculation of the quoted price.

| Criteria, Unit of Measure | Importance, % | Criterion Gradations |
|---|---|---|
| Price, UAH | 80 | - |
| Guarantee period, months | 10 | More than 18 months—10% From 6 to 18 months—5% Less than 6 months—0% |
| Performance time, days | 10 | Less than 30 days—10% From 30 to 60 days—5% More than 60 days—0% |

Source: compiled by the authors according to the (ProZorro infobox 2019).

The criteria for the evaluation of the tender offers and the range (the example is given in Table 4) are indicated in the electronic procurement announcement, and in the tender documents of the tenderer (section "Tender offer evaluation").

**Table 4.** Obtained results of the evaluation of the quoted price under different variants of the tender offer, according to the recommended method (MEDT 2016).

| No | Variants of Tender Offers Conditions | Correction Coefficient | Quoted Price, ths. UAH. | Purchase Price, ths. UAH. |
|---|---|---|---|---|
| 1 | Participant 1 submits an offer for the amount of **10,000** UAH, specifying the performance time of **30** days, the guarantee period of **18** months | 1.19 Calculation: 1 + (0.1 + 0.05)/0.8 | 8403 (10,000/1.19) better offer | 10,000 |
| 2 | Participant 2 submits an offer for the amount of **10,000** UAH, specifying the performance time of **60** days, the guarantee period of **6** months | 1.19 Calculation: 1 + (0.1 + 0.05)/0.8 | 8403 (10,000/1.19) better offer | 10,000 |
| 3 | Participant 3 submits an offer for the amount of **9000** UAH, specifying the performance time of **61** days, the guarantee period of **6** months | 1.06 Calculation: 1 + (0 + 0.05)/0.8 | 8470 (9000/1.06) the worst offer | 9000 |

Source: compiled by the authors.

Legislation of Ukraine "On the Public Procurement" assumes that the customer independently determines the method of evaluation of tender offers, but in fact, except for the method of reducing the quoted price, it is impossible to apply others, as the electronic form of the announcement of the auction does not provide it (Legislation of Ukraine "On the Public Procurement" 2020). The comparative method (Table 5) and the method of valuation (Table 6) are used in world practice. We have carried out calculations of the evaluation of tender offers by different methods, but with the same initial data. The results of the study have shown that the method of reducing the quoted price is not an objective method for choosing the best offer (Table 7).

**Table 5.** Evaluation of the tender offers by the comparative method.

| Participants | Parameters of Evaluation of Tender Offer, Coefficient of Importance | | | | | | | | | Point (Maximum Point is 100) |
| | Price, k = 80% | | | Performance Time, k = 10% | | | Guarantee Period, k = 10% | | | |
| | Price, ths. UAH | $Pn/Pi$ | $k\frac{Pn}{Pi}$ | Calendar days | $Pn/Pi$ | $k\frac{Pn}{Pi}$ | Months | $Pi/Pn$ | $k\frac{Pi}{Pn}$ | |
| 1 | 10,000 | 0.90 | 72 | 30 | 1.00 | 10 | 18 | 1.00 | 10 | 92.00 (the best offer) |
| 2 | 10,000 | 0.90 | 72 | 60 | 0.50 | 5 | 6 | 0.33 | 3.30 | 80.30 (the worst offer) |
| 3 | 9000 | 1.00 | 80 | 61 | 0.49 | 4.9 | 6 | 0.33 | 3.30 | 88.20 |

Source: compiled by the authors.

$$K1 = \sum k\frac{Pi}{Pn}, \tag{4}$$

where

$K1$, $K2$—the total evaluation of the first group criteria;

$k$—the coefficient of importance of the $i$ criterion of the evaluation of the tender offers;

$Pi$—the indicator of the criterion of the $i$ offer;

$Pn$—the indicator of the criterion of the best (base) offer, the second group is the criteria that increase in value, of which reduces the level of competitiveness of the offer (Formula (5)).

$$K2 = \sum K\frac{Pn}{Pi}, \tag{5}$$

where $K2$—the total evaluation of the criteria of the second group.

The essence of the estimated value method (World Bank 2011) is that non-price criteria are transferred in monetary terms regarding the specified minimal requirements. The modifiers that reflect non-price criteria as a percentage of the deviation from the offer price are introduced (Table 6).

The results of the evaluation of the tender offers have shown that the most objective method of the evaluation is the method of comparison, while the method of reducing the quoted price is the best formula by which the worst one can be selected.

The offer of Participant 3 is the worst, according to the source data in Table 7, when using the method of reducing the quoted price, while according to the estimated value method, Participant 3 has the best offer, and he will receive the contract. Participant 1 will receive the contract when using the comparison method. Participant 2 will receive the contract when applying the method of reducing the quoted price, while with other methods, the offer of participant 3 is the worst. As we can see, various methods of the evaluation of the tender offers, in which, in addition to price criteria, non-price

criteria are used, giving different results on the contract award to a tenderer. The method of reducing the quoted price is the only one which is used in Ukraine. The law of Ukraine establishes that the state customer chooses the evaluation method independently, but the ProZorro electronic system is configured only for one method, and excludes the use of other objective methods that are used in international practice.

**Table 6.** Estimated value method.

| Evaluation Criteria | Performance Time | Guarantee Period | Estimated Value |
|---|---|---|---|
| Minimal requirements | 60 days | 6 months | - |
| The modifier of the estimated value of the tender offer | −1% for 10 full days in advance | −1% for increasing the guarantee period on 1 full month | reducing the estimated value on the corresponding% |
| Participant 1 | −2% | −3% | −5% |
| Participant 2 | - | - | - |
| Participant 3 | - | - | - |

Source: compiled by the authors.

**Table 7.** Comparison of the results of the tender offers evaluation by different methods.

| No. | Variants of Tender Offers | Method of Reducing the Quoted Price, UAH | Comparison Method, Point (Maximum Point is 100) | Estimated Value Method, UAH |
|---|---|---|---|---|
| 1 | Participant 1 submits an offer for the amount of 10,000 UAH, specifying the performance time of 30 days, the guarantee period of 18 months | 8403 (the best offer) | 92.00 (the best offer) | 9500 (better offer in comparison with the other) |
| 2 | Participant 2 submits an offer for the amount of 10,000 UAH, specifying the performance time of 60 days, the guarantee period of 6 months | 8403 (the best offer) | 80.30 (the worst offer) | 10,000 (the worst offer) |
| 3 | Participant 3 submits an offer for the amount of 9000 UAH, specifying the performance time of 61 days, the guarantee period of 6 months | 8470 (the worst offer) | 88.20 (better offer in comparison with the other) | 9000 (the best offer) |

Source: compiled by the authors.

Thus, the method of reducing the quoted price, which is used in Ukrainian practice, does not ensure the implementation of the principle of "lower price for better quality", if we compare it with the methods used in international practice.

In March 2020, we conducted a questionnaire survey of state purchasers in Ukraine on the practice of non-price criteria.

There were six questions in the questionnaire. (1) How often do you use non-price criteria to evaluate the tender offers? (2) Do you think that the method of reducing the quoted price allows you to choose the best tenderer? (3) What are the main reasons for your misuse of non-price criteria in public procurements? (4) Do you think that more stringent requirements for the tenderers (for example, mandatory presence of international ISO quality certificates, IOAS certificates of conformity) are an alternative to using non-price criteria? (5) What requirements do you apply to the tenderer in order to reduce the risk of purchasing low-quality products? (6) What actions do you expect from the state regulator of public procurement to enable innovative and green procurements?

Thus, 76% among 92 state purchasers surveyed in March 2020 believed that the method of reducing the quoted price did not allow one to choose the best offer. The respondents said that innovative and

environmentally friendly goods were expensive in comparison with the traditional goods, and the difference in price did not allow the best suppliers to compete, in a situation where unscrupulous suppliers dump during the tenders and win the tenders. Overall, 66% of government purchasers surveyed conspirators did not use non-price criteria, as they believed that they might be a reason for monitoring by the State Audit Office. It should be noted here that the control body (the State Audit Office) does not monitor the criteria for the evaluation of tender proposals. Moreover, 87% of the respondents believed that the application of stricter requirements to suppliers and documentary evidence of compliance was the alternative to non-price criteria. Additionally, 9% of the customers required, from the tenderers, a copy of the certificate of conformity and a copy of the certificate of the accreditation body for conformity assessment (IOAS certificates), in order to reduce the risk of purchasing low-quality products. The choice of a reliable supplier is more important for most customers than the estimation of the total cost of ownership. Furthermore, 91% of the surveyed customers believed that they wanted to see successful practices of the customers who had used non-price criteria to achieve efficiency. Moreover, 98% of the respondents expected, from the regulator, recommendations on the methodology for the assessment of the life cycle value; 11% anticipated recommendations on the method of assessment of the goods impact on the environment; 98% expected recommendations for the purchase of environmentally friendly public procurement.

The recent changes in the EU Public Procurement Directive (Directive 2014/24/EU 2014) have shown a tendency to reject the sole price criterion for the evaluation of the tender offers and the practice of an optimal price-quality ratio is becoming more topical. Models for estimating the total cost of ownership, life cycle cost of public procurement and green public procurement are widely used in the practice of countries with effective public procurement systems. The life cycle cost is calculated at the stage of procurement planning in the international practice, with the use of free online calculators, which are freely available and used as a criterion for the evaluation of the tender offers. In Sweden, online calculators are designed in the form questionnaire for the procurement of lighting, automobiles, vending machines and coffee machines; appliances.

In the construction industry in the EU (the UK, the Netherlands, Portugal, France), LCC contracts (life cycle contracts) are used, under which project payments are made not from the moment of completion of the construction works, but from the moment of the start of the operation of the facility. The payment for the project includes a fee for the service and depends on the quality of execution of the contract terms' requirements. Penalties are imposed on the company in case of non-compliance. The cost of the contract life cycle construction reduces costs to 40% in comparison with budget construction costs.

The G7 countries (Austria, Denmark, Finland, Germany, the Netherlands, Sweden and England) have introduced a public procurement procedure for eco-labeled goods (Nordic Council of Ministers 2017), in order to minimize the negative impact on the environment. The EU directives contain specific recommendations for green procurement:

- inclusion of environmental requirements in the technical conditions;
- use of ecological trademarks;
- application of ecological criteria for the selection of the winner.

The specific subject of green procurement is, for example, paper for recyclable materials printing, the design and construction of an energy-efficient building (Grandia and Voncken 2019; Sönnichsen and Clement 2020). This confirms the fact that corporate social responsibility is becoming an integral part of human development in the world (Činčalová 2017, 2018; Buzko et al. 2019; Trunina and Khovrak 2019; Trynchuk et al. 2019; Činčalová and Hedija 2020).

The qualification requirement for the tenderers should be the experience of performing similar contracts in the subject of procurement. The LCA (life cycle assessment) method is used to evaluate the tender offers, which allows one to determine the most significant impact of the product on the environment, such as minimal $CO_2$ emissions (European Commission 2020). Green public

procurement (GPP) is an increasingly debated "demand side" environmental policy instrument (Brammer and Walker 2011; Cheng et al. 2018; Fuentes-Bargues et al. 2018).

In 2009, Price Waterhouse Coopers (PWC) conducted the research on green public procurement in the G7 countries. The report (The National Agency for Public Procurement 2019) has shown that environmentally sound public procurements can lead to cost reductions over time, rather than cost increases, which is often a common perception.

It is very important to study foreign experience, because in Ukraine, unfortunately, the procurement system does not use such models as: total cost of ownership (TCO) (Ellram 1995), life cycle costing (Hochschorner and Finnveden 2006), and sustainable procurement (Burchard–Dziubińska 2017; Bernal et al. 2019; World Bank 2019).

## 4. Conclusions

The results of the study have shown that the main non-price criteria, which are used by the Ukrainian customers, are the delay of payment term for the delivery of goods (performance of services, works), the experience of suppliers in the market, qualification requirements for the tenderer, the presence of the certificates and licenses. Product quality criteria are used only in 2% of the tenders. Ukrainian customers do not have the modern practice of applying non-price safety criteria, environmental protection measures and innovativeness, and are waiting for recommendations from the regulator. Insignificant and unobjective non-price criteria for the evaluation of the tender offers are often used, especially with the purchase of goods, which leads to a violation of the principle of the maximum efficiency (effectiveness) of procurement. In Ukraine, there are no tools for monitoring non-price criteria, and there are no assessment methods of the effectiveness of the impact of non-price criteria on the efficiency of public procurement. The use of non-price evaluation criteria for the participants in most product groups has led to increased competition in the tenders.

The model of the evaluation of the tender offers with non-price criteria, which is regulated and applied in Ukraine, is unsuccessful for choosing the best offer. The method of reducing the quoted price used in the ProZorro electronic system of public procurement does not provide the best offer, and does not allow innovative suppliers to be competitive, in comparison with unscrupulous suppliers who often dump during the tender. The comparative method is more reasonable with the use of non-price criteria.

According to the survey of the customers on the practice of the use of non-price criteria, it has been determined that the customers want to buy environmentally friendly and innovative products, but have little knowledge of how to apply such criteria properly, and expect, from the regulator, examples of successful practices.

The subject of our further research will be the methodology of the total cost of ownership, the cost of the life cycle of purchases, and green public procurements in terms of their impact on the economic and natural capital of the country, which is a completely new direction for Ukraine.

The influencing factors at the efficiency of public procurement are identified: the objectification of non-price criteria and the method of the evaluation of the tender offers; the presence of the examples of successful procurement with the impact of non-price criteria on the efficiency; the opportunity for the supplier innovators to be competitive in the tenders in the conditions of the application of mass price dumping by the traditional participants; a high level of buyers professionalism, the active efforts of the state regulator in the promotion of the best practice of public procurements.

**Author Contributions:** Conceptualization, A.B., N.T., V.G. and V.L.; methodology, A.B., N.T., V.G. and V.L.; software, N.T., Z.B. and I.Z.; validation, N.T., K.B., Z.B. and I.Z.; formal analysis, A.B., N.T., V.G., V.L., K.B., Z.B. and I.Z.; investigation, A.B., N.T., V.G. and V.L.; resources, A.B., N.T., V.G., V.L., K.B., Z.B., L.B. and I.Z.; data curation, A.B., N.T. and I.Z.; writing—original draft preparation A.B., N.T., V.G., V.L., K.B., Z.B. and I.Z.; writing—review and editing, A.B., N.T., V.G., V.L., L.B.; visualization, N.T., K.B., Z.B. and I.Z.; supervision, A.B. and N.T.; project administration, N.T. and I.Z.; funding acquisition A.B., N.T., V.G., V.L., K.B., Z.B., L.B. and I.Z. All authors have read and agreed to the published version of the manuscript.

**Funding:** This research received no external funding.

**Acknowledgments:** We are grateful to the editorial team for their support and guidance in preparing the publication. Special thanks also go to reviewers who have dedicated their time and experience to evaluating the quality of the submitted manuscript and giving the authors valuable feedback.

**Conflicts of Interest:** The authors declare no conflict of interest.

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
