# Peer review of "Non-Price Criteria for the Evaluation of the Tender Offers in Public Procurement of Ukraine"

_ijfs, doi:10.3390/ijfs8030044_

Round 1
Reviewer 1 Report
Dear authors,
Thank you for engaging in research in such a complex (and many time arid) topic as public procurement.
After reading the article, I´d like to do some suggestions that can improve its quality in my opinion.
- The data and the sample are valid but they are not clearly explained neither organized. The first section is the introduction, after you jump into results and later you come with conclusions right before methods and discussions. I´d suggest to review the organization of the article.
- The writing of the article should be improved. There are mismatches in some verbal tenses, non correct expressions, etc.
- Some references are not correctly cited since you left three points at the end (in example, below table 1 you write (Single...2019).
- The presentation of the results is somewhat confusing, I´d suggest you review it.
- The conclusions are too short, you could include which is your contribution to the academic knowledge and the future lines of research.
- In general, the article needs improvement regarding its presentation and clarification.
Author Response
Dear Reviewer, thank you verymuch for the time and effort that youdedicated to reviewourpaper. We carefully revised the manuscript taking into considerationally our comments and suggestions. Detailed replies to all comments are provided in the following. Kindest regards
- The structure of the article is significantly revised in the sections: Abstract, Introduction, Methodology та Conclusions.
- The article was re-read and formatted properly.
- All references made in accordance with requirements
- The results of the study were linked to the purpose of the study.
- Conclusions expanded for the purpose of research and results.
- The article emphasized the objectivity of the reduced price method, the study of non-price criteria and illustrated the results of the expert survey.
Reviewer 2 Report
The layout of the manuscript is confusing. Your methods section should discuss how you conducted your study prior to any discussion of results/findings. Your methods are unclear and you mention that you're testing hypotheses but those are also not clear. You are missing literature about non-price criteria, such as Drumright, that is older but still informative.
Author Response
Dear Reviewer, thank you for appreciating our submitted work and for providing us helpful comments to improve it.
Kindest regards
authors
We have taken into account your wishes and would like to inform you about changes in the article. The article was structured. Elaborated the literature and supplemented the introduction, methodology, results and conclusions.
Reviewer 3 Report
Dear authors,
I think that the paper is for very accurate interest. It is needed to improve the criteria for public procurement decision, because the price criteria is not efficient for society. In this line I think that the paper fit with the journal and the interest of the literature. However, I will suggest you some aspects to include in your next version after review the paper:
- It need to establish what is the mean of non-price criteria, because I thought that it was a very different aspect comparing with price, but after read the paper it is the adjusted price. Then, I cannot understand your proposal.
- Link with the previous one: What is the different between Pa – the adjusted price and Price? Most of included variables are based on price, and then I am very confused about your suggestion and contribution of this paper.
It is link to the risk, but are other aspects in positive relevant to the price criterion? I value very positively, but the perspective is correct the risk to negative aspects on established price. I think that it is a great progress; however, I would like to read your opinion, or at least for future research your ideas about other positive aspects to include as decision criterion. (see the link to some of our ideas if you want)
- Why you have chosen Ukraine, the law is different comparing with EU Countries?
- You have included three aspects: costs, quality and level of service. We have analyzed using a Delphi that those aspects you mentioned are important; however, we concluded that not only those aspects are important, but also some other relate to the value generation of companies (tenders). It is important because of the quality of the services (not only qualitative sense, but also in a quantitative way). Moreover, some aspects relate to employment, satisfaction or territory development will be for the interest of the public procurement decisions. The link to the article is: https://www.mdpi.com/2071-1050/11/15/4069
- How do you measure the quality? Because in some table (Table 5) I have read time. Is it the same or linked?
- Aims and Methods section: include some table with sum of the ProZorre, some key aspects and relevancy. If not, delete it or include in other section (in this version is very short)
- Conclusions: You do not suggest any alternative. Then, I recommend in-depth argument in this section around implications. In addition, it is need to include the contribution to the theory.
- Include limitations and future research (own section or include in Conclusion section)
- References: check and include some Sustainability special issues papers for example around public procurement non-based on price or other journals papers are welcome, as well, of course.
- Some Annex with used data/examples will be welcome.
Good luck
Author Response
Dear Reviewer, we thank you very much for you refforts and the review of our article.
Kindest regards
authors
Withregard to yourcomments, pleaseallowus to explain as follows:
- Non-price criteria are criteria that are not measured in monetary units, but in combination with price criteria are able to take them into account.
- The reduced price method is the only method that Ukrainian customers can use when applying non-price criteria. In our opinion, this method is not objective, our calculations and the results of the survey confirmed it.
- Ukraine was chosen because the practice of applying non-price criteria is very different from European practice.
- We have removed this aspect from the article.
- We have removed this aspect from the article. We will thoroughly investigate the issue of criteria in the future, so we decided to refine it and not submit it in this article.
- The section has been revised and supplemented.
- In our opinion, this issue is debatable. To ensure the effectiveness of public procurement, government officials need the following tools: objectification of non-price criteria and the method of evaluation of tender proposals; the presence of examples of successful procurement with the impact of non-price criteria on the efficiency; the opportunity for the suppliers innovators to be competitive in the tenders in the conditions of the application of mass price dumping by the traditional participants; high level of buyers professionalism.
- In the conclusions it was noted that the criteria for evaluating tender proposals will be the subject of our further research.
- We have supplemented the list of references
- The study of non-price criteria was illustrated by the results of an expert survey of customers.
Round 2
Reviewer 1 Report
Dear authors,
Thank you for resending a much improved version of the article. Nevertheless, there are still some aspects that would need improvement in my opinion.
- The English language have some mistakes in the form of verbal tenses, etc. For instance, in page tender organizers has gave instead of tender organizares have given. The article would benefit from an extensive English proof reading.
- Some of the citations are incorrect. for instance (Indicators (2018-2019) 2019) or (Single Market Scoreboard 2019). This last one belongs to the European Commission, and its not a source itseld
- The conclusion has improved a lot. Still there are some "orphan" lines like "The criteria for evaluating tender proposals will be the subject of our further research"
- The results have also improved but I still miss a more detailed explanation of the comparisons you made. The different methods used are still blurry, meaning that it is necessary in my opinion to enlarge the explanation.
- The text after the (ex) figure 3 does not quite fit on the results part. In addition, you refer to surveys made to Ukrainian costumers, where those surveys performed by you of from an external source? It is not said in the methodology, neither in the results part. Shall the first be the case (own elaboration) I´d suggest you explain the process.
Hope these comments will help you to improve the article,
Kind Regards
Author Response
Dear Reviewer, thank you for appreciating our submitted work and for providing us helpful comments to improve it.
- The text has been read again and the errors have been corrected.
- Quotes have been redesigned.
Bi.prozorro. 2019. Indicators (2018-2019). Available online: http://bi.prozorro.org/http/sense/app/fba3f2f2-cf55-40a0-a79f-b74f5ce947c2/sheet/HbXjQep/state/analysis#view/pEh (accessed 20 january 2020).
Crown Agents. 2017a. Public procurement guidelines. The EU Project: Harmonisation of Public Procurement System in Ukraine with EU Standards. Available online: https://eupublicprocurement.org.ua/wp-content/uploads/2017/10/Guidelines_UKR_interactive_pages.pdf (accessed 12 january 2020).
Crown Agents. 2017b. Harmonization of public procurement system in Ukraine with EU standards. Guidelines on public procurement award criteria on best price/quality ratio. Available online: https://translate.google.com/translate?hl=ru&sl=en&u=http://eupublicprocurement.org.ua/wp-content/uploads/2017/09/Guidelines_on-award-criteria_ENG.pdf&prev=search (accessed 10 march 2020).
European Commission. 2019. Single Market Scoreboard. Public Procurement Reporting period: 01/2018–12/2018. Available online: https://ec.europa.eu/internal_market/scoreboard/performance_per_policy_area/public_procurement/index_en.htm (accessed 23 may 2019).
European Commission. 2020. Green Public Procurement. Available online: http://ec.europa.eu/environment/gpp/index_en.htm (accessed 15 march 2020).
Nordic Council of Ministers. 2017. Nordic guidelines – green public procurement. How to use environmental management systems and ecolabels in EU tenders. Available online: http://norden.diva-portal.org/smash/get/diva2:1087097/FULLTEXT01.pdf (accessed 15 june 2019).
ProZorro infobox. 2019. Non-price criteria of evaluation. Available online: https://infobox.prozorro.org/articles/necinovi-kriteriji-ocinki (accessed 7 march 2020).
ProZorro. 2018. Construction works and repair (works of an unfinished building for service housing for servicemen). Available online: https://prozorro.gov.ua/tender/UA-2018-05-23-000763-b?fbclid=IwAR0hoxAnxQQ8C8g3h6r22opvfP7ZRbs1NCayXKVei3COJO4D1dQWgsEMY54 (accessed 10 march 2020).
- The comments have been taken into account. It was added paragraph:
The subject of our further research will be the methodology of the total cost of ownership, the cost of the life cycle of purchases, green public procurements in terms of their impact on the economic and natural capital of the country, which is a completely new direction for Ukraine.
- The article has been added an explanation: Using the method of reducing the present price the worst offer, according to the source data of the table. 7, is the offer of participant 3, while according to the estimated cost method, participant 3 has the best offer and he will receive the contract. Using the comparison method, participant 1 will receive the contract. Participant 2 will receive the contract when applying the reduced price method, while with other methods the offer of participant 3 is the worst. As we see, various methods of evaluating tender offers, in which, in addition to price criteria, non-price are used, give different results on the award of a contract to a bidder. In Ukraine, only one method is used - the method when the price decreases. The law of Ukraine establishes that the state customer independently chooses the assessment method, but the ProZorro electronic system is configured only for one method and excludes the use of other methods that are used in international practice and are more objective.
- The text was structured, the methodology process was explained in paragraph: The questionnaire contained the following questions: how often do you use non-price criteria to evaluate tenders? Do you think that the reduced price method allows you to choose the best bidder? What are the main reasons for your non-use of non-price criteria in public procurement? Do you think that more stringent requirements for bidders (for example, such as the mandatory presence of international ISO quality certificates, IOAS certificates of conformity) are an alternative to using non-price criteria? do you apply the requirements to the bidder in order to reduce the risk of receiving low-quality products? What actions do you expect from the state regulator of public procurement to enable innovative and green procurements?
Sincerely, Authors
Reviewer 2 Report
I have no comments
Author Response
Dear Reviewer, thank you for appreciating our submitted work and for providing us helpful comments to improve it.
Dear Reviewer, thank you for appreciating our submitted work and for providing us helpful comments to improve it.
The text has been read again and the errors have been corrected.
Sincerely, Authors
Round 3
Reviewer 1 Report
Dear authors,
Thank you for the extent revision you have performed. The article has much improved from the original version and now it only needs some review on the redaction and it will be ready to publish.
Kind Regards,
Javier